# Research on the Influence of Processing Parameters on the Specific Tensile Strength of FDM Additive Manufactured PET-G and PLA Materials

**DOI:** 10.3390/polym14122446

**Published:** 2022-06-16

**Authors:** Michał Bembenek, Łukasz Kowalski, Agnieszka Kosoń-Schab

**Affiliations:** Faculty of Mechanical Engineering and Robotics, AGH University of Science and Technology, A. Mickiewicza 30, 30-059 Kraków, Poland; lkowalski@agh.edu.pl (Ł.K.); koson@agh.edu.pl (A.K.-S.)

**Keywords:** FDM, PLA, PET-G, tensile strength, additive manufacturing, 3D printing

## Abstract

Fused deposition modeling (FDM) is one of the most accessible additive manufacturing (AM) technologies for processing polymeric materials. It allows processing most of thermoplastic polymers, with polyethylene terephthalate glycol-modified (PET-G) and polylactic acid (PLA). AM parts tend to display anisotropic behavior because of layer-by-layer fabrication and various technological parameters that can be set for 3D print, so it is hard to predict and analyze how the manufactured parts would behave under load. This research presents results of classic tensile strength tests performed on 57 PET-G specimens and 57 PLA specimens manufactured with varying technological parameters such as: printing temperature, print orientation, layer height, and infill percentage. Afterward, a comparative analysis is performed, proposing specific tensile strength (STS) as a benchmark to determine how 3D printed parts strength is varying due to beforementioned parameters, eliminating bias induced by varying weight of specimens. The biggest relative increase of UTS and the biggest relative decrease of STS was noted for variable infill percentage (increasing infill—PLA: 37.27% UTS increase and 30.41% STS decrease; PET-G: 24.42% UTS increase and 37.69% STS decrease). The biggest relative increase of STS between examined parameters was observed for both materials as the printing temperature was increased (27.53% for PLA and 12.69% for PET-G). Similar trends in STS changes were observed for both materials. Obtained data shows which FDM AM parameters are the most important to obtain the biggest UTS of manufactured parts, and those do not overlap with parameters needed to obtain optimal strength-to-weight ratio.

## 1. Introduction

3D printing, also known as spatial printing, involves producing a physical object seen in three dimensions by depositing thin layers of material stacked on top of each other [1]. 3D printing is seen as a modern tool used in industrial prototyping processes. This technology is also called additive or generative [2]. These technologies are used in many functional areas of every industrial field. For several years, additive technologies have been dynamically developing processes of industrial serial production. This trend is particularly visible in the aviation, space industry, and locomotive industries [3,4,5]. This technology is playing an increasingly important role in clinical practice. In the field of surgery, 3D printed models have proven useful in surgical planning [6], medical education [7], medical consultation, organ transplantation, plastic surgery, and prosthesis implantation [8,9]. Additive manufacturing (AM) is rapidly evolving into a comprehensive approach to manufacturing, facilitating the design and enabling prototyping and mock-up technologies [10]. It is one of the methods allowing for a quick achievement of precise shapes of products and is characterized by a high rate of material utilization [11]. Previous research on materials used for 3D printing has led to the development of polymers with favorable mechanical properties and biocompatibility, where mechanical properties can be tuned by changing the parameters of the printing process [12]. 

The most commonly used printing method is fused deposition modeling (FDM) [13,14]; unfortunately, due to several drawbacks such as low surface quality, poor mechanical properties, and long fabrication time, applications are severely limited [15]. To improve the quality of prints by this method, the effect of various process parameters such as layer thickness, molding direction, filling rate, filling density [16], and printing angles on mechanical properties including tensile strength and elastic properties of materials used in additive manufacturing methods of FDM [11]. 

The most popular material used for FDM printing is polylactic acid (PLA) [17]. Scientific experiments have shown that during the molding process, the PLA material undergoes a melting and solidification process, due to thermal shrinkage, which causes a deterioration of the mechanical properties [18]. The change infill density mainly determines the tensile strength. It has been proved that the spatial orientation of 3D printing with PLA has less effect on Young’s modulus and more on tensile strength. Increasing the number of layers leads to a decrease in both Young’s modulus and tensile strength [17]. The effects of layer height and plate temperature on the impact strength of printed PLA were investigated. It was shown that an appropriate choice of these parameters influences increasing the impact strength of the printed sample [19]. The topic of compressive Young’s modulus of PLA additively manufactured parts was discussed in [20], where authors concluded that test load orientation on cubical 3D printed specimen affects analyzed parameters differently, depending on the type of PLA material used, while using different cross-section shapes influences the test results minimally.

A theoretical model was developed to predict the tensile strength of FDM PLA materials based on the transverse isotropy hypothesis, classical lamination theory, and Hill–Tsai anisotropic plasticity criterion, which was verified by tensile experiments. The theoretical model and experimental method can also be applied to other 3D printing materials produced by FDM or SLA (stereolitography) techniques [21,22]. In this paper, Zhao, Chen, and Zhou presented two new theoretical models for predicting the tensile strength and Young’s modulus of PLA material fabricated by additive FDM method with different printing angles and layer thicknesses. These models were developed based on the hypothesis of transversely isotropic material and Tsai–Hill strength criterion and the hypothesis of orthotropic material in-plane stress state [23].

Moradi et al. showed that decreasing the film thickness at the same printing speed increases the cooling rate. The effect of such measures is to increase the strength and decrease the elongation of the test specimens. Increasing the number of contour layers from 2 to 6 increases the maximum failure force by 42%. Increasing the number of contour layers, due to an effect similar to increasing the fill density, increases the breaking force and production time [24,25].

In the available literature, one can find studies devoted to PET-G. Szykiedans et al., basing on ISO 527 conducted mechanical strength tests on samples with different thicknesses. Their study showed that the 3D printed PET-G samples are anisotropic but have different tensile modulus values due to the presence of air gaps in the printed structure and stress concentration along with the fiber beads [26].

In the paper [27], tensile test specimens were fabricated at the same printing speed (45 mm/s), extruder temperature (240 °C), table temperature (70 °C), and layer thickness with different occupancy levels (20%, 50%, 80%) for experimental studies. The effect of print occupancy on the mechanical properties and surface roughness of the PET-G products was investigated. Measurements of uniaxial tensile test, hardness, and surface roughness of the specimens showed that as the occupancy level increases, the average tensile strength values increase while the average elongation values decrease. This does not always lead to a similar increase in surface roughness values.

An attempt was made to compare polylactic acid (PLA) and polyethylene terephthalate glycol (PETG) materials. Samples produced by FDM technology at different printing temperature and printing speed were subjected to tensile, compressive, bending, and thermal deformation tests.

The results showed that the PLA and PETG materials exhibit a distinct asymmetry in tension and compression. The mechanical properties (tension, compression, and bending) of PLA and PETG increase at higher printing temperatures, and the effect of velocity on PLA and PETG does not show consistent results. The mechanical properties of PLA have higher values than those of PETG, but the thermal deformation for PLA samples is lower [28]. Many tests have been performed to determine the technological and strength properties of the materials used for printing. Tests were conducted on various shapes of samples with different print orientations [29]. Abbot and Kallon used 25 mm × 25 mm × 25 mm block-shaped specimens previously designed using Autodesk Inventor 2018 and tested in the Autodesk Inventor simulation environment to gain insight into the response of these objects under compressive loading. The specimens were then fabricated according to the design using a 3D printer with several different materials and fillings and loaded according to the simulation, which allowed comparison with the finite element method results. The purpose of this study was to identify the best process used for personal printers in terms of cost, durability, surface roughness, and human perception [30]. The analysis and study were also applied to lattice-shaped samples with different layer thicknesses but the same mass. Li et al. designed and fabricated new T-rib hierarchical panels with excellent mechanical properties by 3D printing technique. Their mechanical properties were investigated by compression tests, FEM, and theoretical analyses. From the study, it was found that the hierarchical structure can simultaneously increase the global bending stiffness and local bending stiffness to prevent instability and local buckling [31]. 

The direction of printing is a key parameter affecting the rheological properties of materials. Analyzing the results of the study performed by Kozior and Kundera, it can be concluded that the highest values of Young’s modulus are achieved by specimens made at the specimen printing angle of 0° (cylindrical specimen’s axis of rotation is parallel to the printer’s buildplate). Rheological properties indicate that the 0° position of the specimen results in the highest modulus of elasticity E and E1 in the Standard I rheological model. The 45° position of the specimen (sample’s axis of rotation is at 45° angle to the buildplate) results in the lowest modulus of elasticity for constant loads and the lowest values of all the moduli mentioned above [32].

By analyzing the state of environmental pollution from polymer waste, a comparative study of the mechanical properties of material samples 3D printed from virgin polylactic with those printed from recycled PLA was undertaken. The Taguchi method was used to investigate the effects of FDM manufacturing parameters on the tensile strength, three-point bending strength, and impact strength of 3D printed test parts made of PLA and recycled polylactic acid (Re-PLA), analyzingprocessing parameters such asthree different layer thicknesses (0.15, 0.20, and 0.25 mm) and filling degree (30%, 50%, and 70%),with rectilinear filling structure [3]. The recycled samples showed similar properties to the virgin samples, with a slight decrease in most areas. All differences between virgin and recycled material were significant (values *p* < 0.005), except for the tensile modulus [33]. There were difficulties in printing with recycled fiber, which sometimes clogged the printer nozzle [34]. 

The topic of additively manufactured parts’ physical properties, as presented above is widely researched in recent years. The strength of materials is an important physical property of FDM 3D print, especially because the technology is used to make functional parts of various devices’ prototypes, as well as parts for devices and machines manufactured in small batches. Recent developments in the field focus mainly on determining tensile strength of FDM 3D printed materials with a focus on specific thermoplastic material types. A model allowing to predict mechanical behavior of 3D prints is yet to be created. 

The goal of this research is to present a comparative research approach that allows to determine similarities between different materials mechanical behavior. Doing so allows to find how much different processing parameters affect tensile strength, while indirectly excluding various material-specific parametersand determine which processing and manufacturing parameters are less important for part’s strength. Selecting STS as a basis for comparison helps to remove parameters such as flowability and true density of the raw material from the data obtained. Another goal was to use the method to achieve data helping to improve it in the future by finding out the level of dissimilarities between materials for further analysis. Acquired data is the first step in optimizing the manufacturing process, from base material through printing parameters to mechanical part properties and strength-to-weight ratio. It is important in making functional mechanical parts, as well as for improving economical affordability in midscale manufacturing.

## 2. Materials and Methods

Two types of polymeric materials were chosen for the experiment—polylactic acid (PLA, yellow dye) and polyethylene terephthalate glycol-modified (PET-G, brown dye), both manufactured by Spectrum Group (Pecice, Poland). Shape of the specimens was selected to be similar to the ones suggested by ASTM D638 standard for tensile properties of plastics (Figure 1). 

The specimens were prepared using Creality Ender 3 v.2 (Shenzen, China) FDM 3D printer in sets of 3 pieces, with printing parameters for specific sets presented in Table 1. G-code instructions for the printer were prepared using Ultimaker Cura (Ultimaker, Utrecht, Netherlands) software. Each sample was manufactured separately, located on the same print-area coordinates with the exception of sample sets numbered 17–19. Additional parameters constant for all of the samples were: external wall thickness—1.2 mm, amount of top and bottom layers—5, infill printing speed—60 mm/s, outlines (external walls) printing speed—30 mm/s, extruder nozzle diameter—0.4 mm.

The variable parameters for both materials were: temperature (sets 1–5), infill type (sets 6–8), infill percentage (sets 9–13), layer height (sets 14–16), and sample orientation on 3D printer’s build plate (sets 17–19). For clarification, the build plate orientation is presented on Figure 2, where XY plane represents the build plate. On Figure 3 the examined infill patterns are presented, with trial-default 25% infill percentage. In YZ orientation, a small number of support structures were used in the middle of the overhang on the side of the specimen.

Total number of test samples prepared amounted to 57 pieces for PLA and 57 pieces for PET-G material. After the manufacturing process, the specimens were weighed using Salter 1260 SVDR portable scale (Salter Housewares Ltd., Manchester, UK) with measurement accuracy being 0.05 g. After that, all of the test specimens underwent quasistatic tensile strength tests using Instron 4500 Series Universal Testing Machine (Instron, Norwood, MA, USA) with the testing frame capable of applying maximal load of 10 kN. The trials were performed with constant elongation speed of 5 mm/s until material failure with 20 Hz sampling rate. The test results were recorded and processed using MS Excel 2010 (Microsoft, Redmond, WA, USA) software. The ultimate tensile strength of the material (UTS) was determined as maximal applied force to initial cross section area ratio (Equation (1)). The initial cross section area of the specimen was calculated separately for each specimen before the test using manual measurement data, with simplification that the cross section is a solid block.
(1)$$R_{m}=\sigma_{max}=\frac{F_{max}}{S}\left(MPa\right)$$
where: *R_m_*—tensile strength of a specimen in *MPa*, *F_max_*—maximal force recorded in N, *S*—specimen initial cross-section area in mm^2^.

Afterward, the specific tensile strength (STS) was calculated for each specimen by dividing UTS by the sample density (calculated using computer-aided design software) (Equations (2) and (3)).
(2)$$R_{m_{specific}}=\frac{R_{m}}{\rho }\left(\frac{Nm}{kg}\right)$$
(3)$$\rho =\frac{m_{measured}}{V}{\left(\frac{kg}{m^{3}}\right)}_{}$$
where: *R_m specific_*—specific tensile strength, in *Nm/kg*, *ρ*—calculated material density, in kg/m^3^, *m_measured_*—measured weight of a sample, in *kg*, *V*—volume of a sample calculated by Inventor CAD software (Autodesk Inc., San Rafael, CA, USA), in *m*^3^.

Resultant STS was then used as a benchmark value to determine how analyzed additive manufacturing parameters change material strength properties without bias that is introduced by material density fluctuations. For each of the specimens sets, arithmetic mean for the analyzed and measured properties was calculated.

## 3. Results and Discussion

### 3.1. PET-G Results

Measured and calculated properties for PET-G additive manufactured samples are presented in Table 2. Calculated standard deviation for samples’ sets weight and UTS measurements is presented in the table in parenthesis after the value average. It is visible that slopes for UTS do not match respective slopes of STS. The parameter displaying the highest UTS change between the highest and the lowest average value is infill percentage, where UTS increase of 24.42% can be observed. For the same manufacturing variable parameter, however, while analyzing STS, decrease of value is visible, showing the largest decline of 47.95% between the strongest and the weakest specimen set. In conclusion, increasing infill percentage leads to the least beneficial results. For specimens printed with temperature as variable, 5.72% weight increase can be observed (between set 1 and 5, 210 and 250 °C). As visible on Figure 4, this parameter caused the largest increase of specimen weight. 

Gradual increase between variable temperature sets was also noted, being 16.86% and 12.7% for UTS and STS, respectively. This is the largest STS increase between all variable parameters examined (excluding the orientation variable, which cannot be compared to others due to structural differences of AM sample). In conclusion, selecting proper extruder temperature is crucial to obtain optimal weight-to-strength ratio of a FDM 3D printed part.

The largest specimen relative weight increase was noted for samples’ sets 14–16 (variable layer height), specifically 17.53% relative increase between sets 14 and 16. For this variable, both UTS and STS show appreciation trends, similarly to variable temperature sets. The percentage difference here for STS is the smallest (2.53%), and for UTS, the second largest, equal to 19.61%. The difference can be explained by another fixed building parameter, specifically number of top and bottom (external) layers with 100% infill. For set 14 (0.12 mm layer height), the top/bottom fully filled surface thickness amounts to 0.6 mm, and for set 16 (0.28 layer height)—1.4 mm.

While data about variable layer height and variable infill type cannot be compared directly to other results, they also give useful information about how specimens’ behavior changes with alterations to its internal structure. For variable orientation sets, the XZ samples displayed the lowest STS and UTS (in comparison to XY and YZ orientations), and the highest for XY. Percentage difference between means for XY and XY were 13.25% for UTS and 29.29% for STS—almost double the difference between beforementioned percentages show that proper print orientation affects weight-to-strength ratio more than just raw tensile strength. In the working printed models case, however, with lower material usage during printing, the amount of support structures required for proper model printing should be considered when deciding on its orientation on the printer’s buildplate.

Variable infill types specimens’ sets (numbered 6–8) display percentage change between lines infill (highest) and triangle infill (lowest) of 7.75% for UTS and 7.11% for STS. Comparing STS to UTS values for different infills shows that selecting the right infill for the process can increase both to comparable degree.

### 3.2. PLA Results and Comparison

Same parameters’ changes were analyzed and calculated for PLA material. Experiments results are presented in Table 3. For variable infill percentage sample sets (9–13) obtained results displayed the same trend for UTS as PET-G (Figure 5)— increase of infill percentage value led to increase of tensile strength, with maximal percentage difference between sets 9 and 13 equal to 30.41% and, respectively, weight increase of 52.57% (corresponding samples made from PET-G increased their weight by 52.9%). However, the STS for those sets reached minimal value at 50% infill instead of 100%, although the decrease of STS was still visible. Infill level of 0% still yielded the highest STS, so it is optimal to use lower infill percentage, with the amount only allowing to print external shells properly (without optically visible errors). Graphical representation of UTS and STS trend lines are visible on Figure 6.

Simplifying the data interpretation by assuming linear change as the infill percentage increases, it is possible to extrapolate optimal strength-to-weight ratio in both materials’ cases using obtained linear approximation formulas presented there. It is visible that for both materials, the optimal value is close to 50% infill.

Variable temperature for PLA sets achieved higher relative increase of both STS and UTS than those made from PET-G (UTS: 27.52% and STS: 34.5% between sets 1 and 5), leading to the conclusion that the range of temperatures used for AM from PLA material should be smaller, or the processing window adjusted in future research. For standardization purposes, the range of temperatures for testing between different materials should be adjusted in a manner that allows obtaining similar strength appreciation speed for both (or multiple) tested materials (those ranges do not have to overlap).

We still appreciate the trend for STS is the strongest between tested variable parameters, as is the same with the PET-G samples.

For samples made of PLA, variable layer height yielded the best UTS results with 0.28 mm height, but STS decreased with growing layer height, as opposed to increase for PET-G, which was discussed earlier. This difference might be caused by three properties—increase of top and bottom shell thickness (same as for PET-G), different raw material density, and different layer bonding strength. The differences in raw material density can be caused by inclusions of micro air bubbles in PET-G samples during printing, as by average both materials in raw form should have the same density. It is visible in weight relative increase between specimens sets 14 and 16: 17.52% for PET-G and 25.43% for PLA. Influence of layer bonding strength should be visible in UTS relative change; however, 18.4% change for PLA and 19.61% change for PET-G (between sets 14 and 16) are smaller than weight ratios, leading to conclusion that the main parameter influencing strength in analyzed case is raw material density. 

PLA samples that were printed with altering orientation on build plate (sets 17–19) had the highest STS and UTS for XY orientation, and lowest for XZ, which was also the case for PET-G specimens. Mean weights difference between PLA XY and XZ sets was equal to 5.02% and for respective PET-G sets: 4.73%. Differences in UTS and STS for the same sets were larger than for their PET-G counterparts (UTS: 55.99% and STS 58.21%). This can be caused by raw materials densities differences and layer bonding strength, similarly as for variable layer height sets, but because of the previously mentioned weight ratios and small differences between them, in future research, influence of layer bonding strength on UTS and STS for varying specimen orientation on the buildplate should be analyzed. Similarly, as for PET-G, PLA additively manufactured parts’ STS increases faster than UTS with space orientation change. 

Concerning infill types, PLA yielded different results than PET-G—the highest STS and UTS was measured for cubic infill, and the lowest for lines infill (which was the strongest infill type for PET-G specimens). Due to this observation, future research can be conducted to determine causes of this phenomenon, possibly, by analyzing the elastoplastic behavior of both materials (elongation of PLA in elastic range of deformation for PLA is smaller than for PET-G). The differences are visible on Figure 7, which represents the test data obtained from specimen number 7 (sample set 3)—midrange processing temperature and triangles infill type.

For these process parameters, elastic range of PLA is minimal, while for PET-G, it is visible.

## 4. Conclusions

Specific tensile strength of 3D printable materials can be used to quickly evaluate optimal weight-to-strength ratio. Similar material behavior was observed for PET-G and PLA while measuring UTS changes for varying manufacturing parameters such as: hot-end temperature, infill percentage, and orientation on the printer’s buildplate. Similar behavior was observed for STS measurements on both materials for variable parameters such as: hot-end temperature and orientation on the printer’s buildplate. Those observations lead to conclusion that beforementioned parameters changes with correlation to STS and UTS can be universal to more 3D printed materials, which can be further investigated in future research.

UTS displays reverse trend to STS as infill percentage increases—the more infill, the less optimal weight-to-strength ratio is (for both materials). Percentage changes of UTS and STS for both materials can be presented in decreasing order (to present the most and the least important processing parameters). The order for UTS is: spatial orientation->infill percentage->hot-end temperature; for STS (PET-G): infill percentage->spatial orientation->hot-end temperature; and for STS(PLA): spatial orientation->infill percentage->hot-end temperature. Since percentage changes of specimens’ weights between PLA and PET-G were small (for variable spatial orientation samples, as mentioned in results discussion) and said specimens presented the largest change between themselves for PLA exclusively, further research on this phenomena should be conducted, possibly by analyzing layer bond’s strength influence on STS as well as the influence of the elastoplastic behavior of material (elongation before cracking during tensile test) on various materials’ strength.

Similar trend in relation to layer height change was observed for UTS measurements in both materials but varied for STS. By analyzing the weight change between samples with 0.12 mm and 0.28 mm layer thicknesses and knowing that density of raw material should be the same for PLA and PET-G, foaming susceptibility for different filaments and its influence on AM parts strength can be investigated in future research and introduced to 3D prints’ strength models.

Change in infill type affected both materials differently: cubic infill yielded best results (UTS and STS) for PLA, and for PET-G, calculated values of analyzed strength parameters were the highest for lines infill. Causes of those differences can be investigated in more detail, by researching differences between deformation processes for various AM materials and comparing parameters such as elastic modulus and Poisson’s ratio.

## Figures and Tables

**Figure 1 polymers-14-02446-f001:**
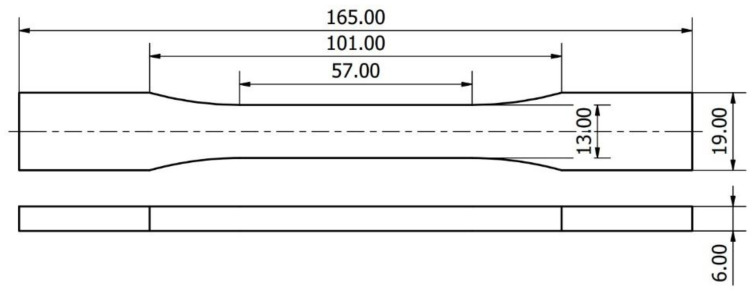
Specimens’ dimensions and shape.

**Figure 2 polymers-14-02446-f002:**
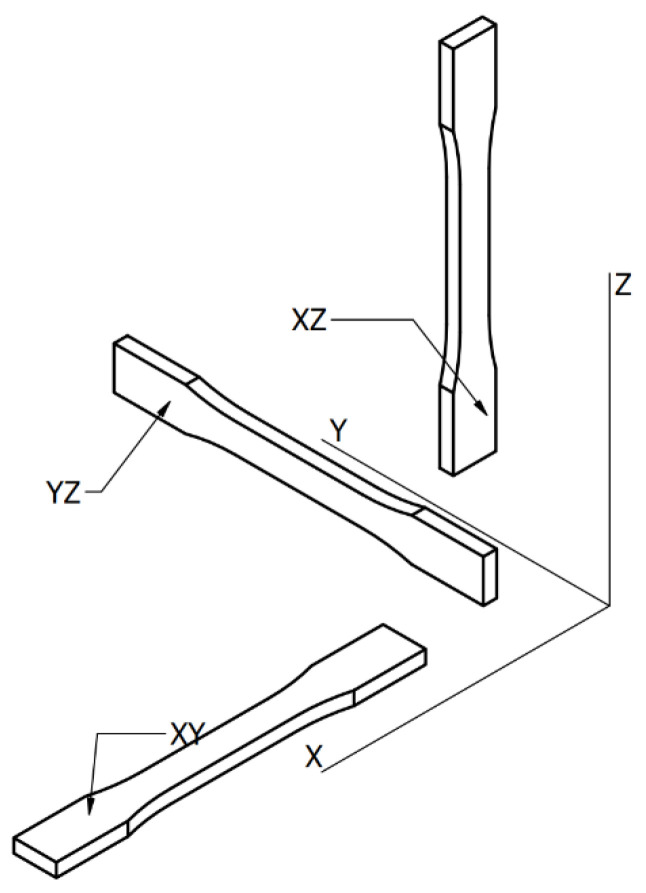
Specimen build plate orientation.

**Figure 3 polymers-14-02446-f003:**
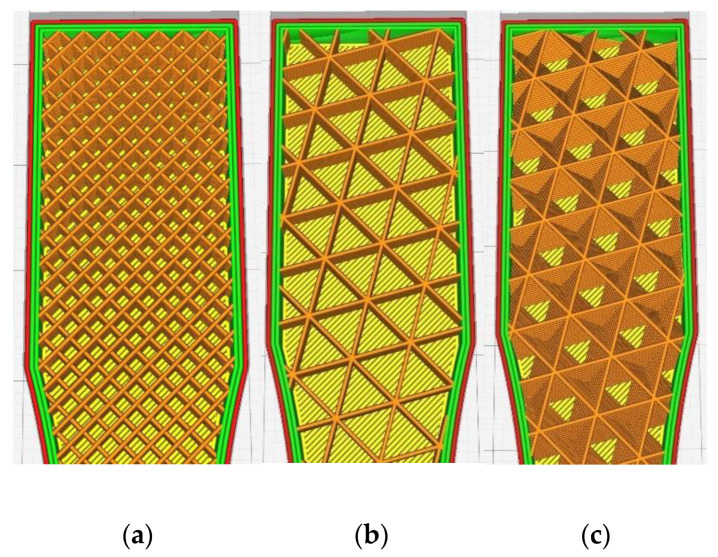
Examined infill patterns: (**a**) lines, (**b**) triangles, and (**c**) cubic.

**Figure 4 polymers-14-02446-f004:**
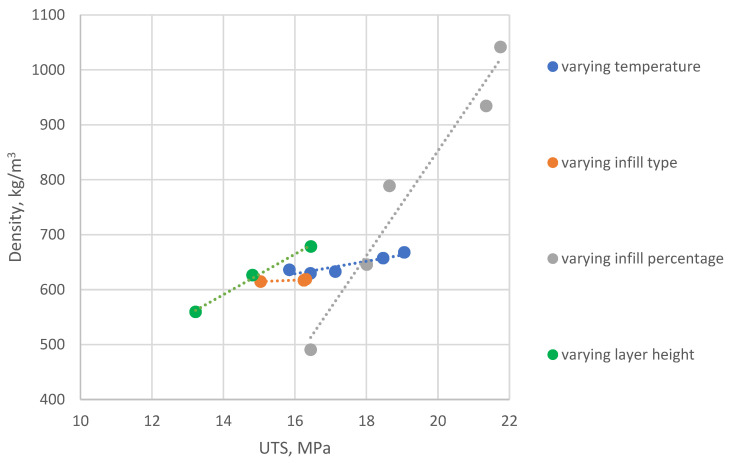
Graphical representation of UTS vs. specimens’ density measured data (PET-G): varying temperature—sets No. 1–5; varying infill type—sets No. 6–8; varying infill percentage—sets No. 9–13; varying layer height—sets No. 14–16.

**Figure 5 polymers-14-02446-f005:**
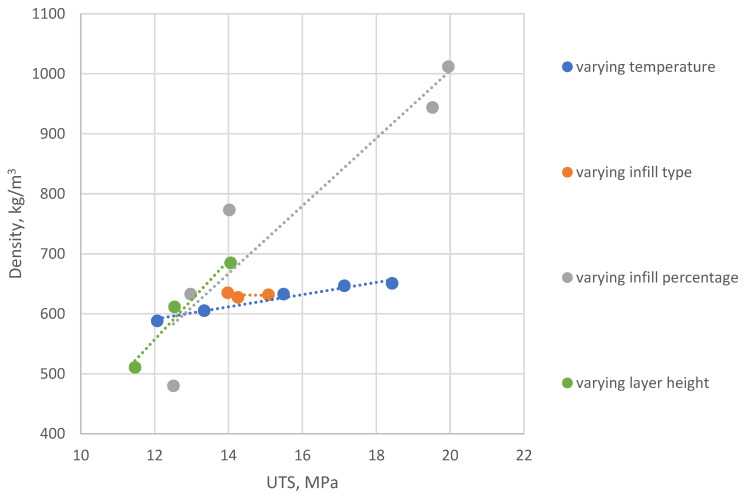
Graphical representation of UTS vs. specimens’ density measured data (PLA): varying temperature—sets No. 1–5; varying infill type—sets No. 6–8; varying infill percentage—sets No. 9–13; varying layer height—sets No. 14–16.

**Figure 6 polymers-14-02446-f006:**
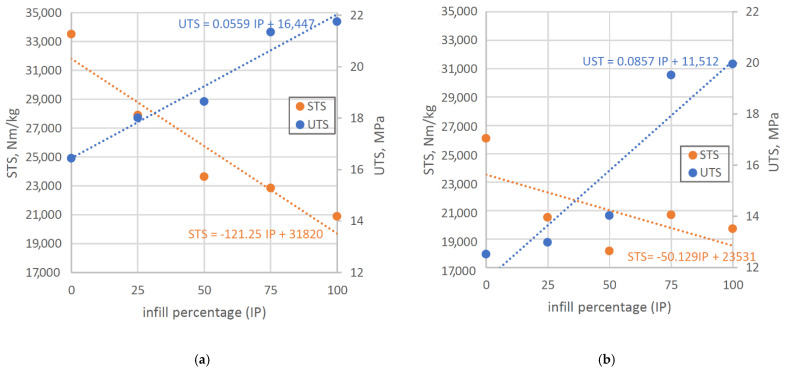
UTS/STS vs. infill percentage graphs for PET-G (**a**) and PLA (**b**).

**Figure 7 polymers-14-02446-f007:**
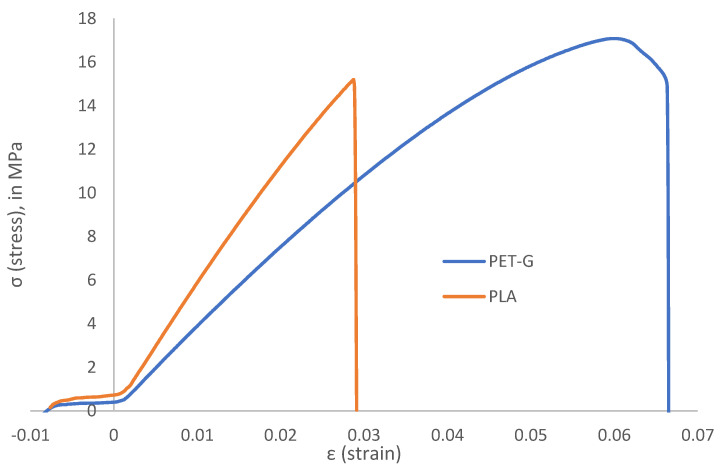
Comparison between PLA and PET-G raw elongation data for specimen 7 (set 3).

**Table 1 polymers-14-02446-t001:** Specimens’ manufacturing parameters.

Number of Set	Temperature (PET-G/PLA), °C	Infill, %	Infill Type	Layer Height (mm)	Printing Orientation
1	210/180	25	triangle	0.20	XY
2	220/190				
3	230/200				
4	240/210				
5	250/220				
6	230/200	25	cubic	0.20	XY
7			line		
8			triangle		
9		0	line	0.20	XY
10		25			
11	230/200	50			
12		75			
13		100			
14				0.12	XY
15	230/200	25	triangle	0.20	
16				0.28	
17	230/200	25	triangle	0.2	XY
18					XZ
19					YZ

**Table 2 polymers-14-02446-t002:** Test results for PET-G material.

Variable Parameter	Set Number (as in Table 1)	Mean Weight of Set, kg (st. dev.)	Mean UTS, MPa (st. dev.)	Mean Density of Set, kg/m^3^	Mean STS of Set, Nm/kg
temperature	1	0.00998(0.00011)	15.844(0.22)	635.96	24,914.1
	2	0.00988(5.82× 10^−5^)	16.436(0.15)	629.59	26,105.9
	3	0.00993(2.20 × 10^−5^)	17.130(0.20)	632.77	27,071.8
	4	0.01032(0.00012)	18.468(0.25)	657.19	28,100.8
	5	0.01048(4.4 × 10^−5^)	19.058(0.17)	667.81	28,538.1
infill type	6	0.00968(5.82 × 10^−5^)	16.251(1.21)	616.85	26,345.7
	7	0.00972(2.20 × 10^−5^)	16.306(0.40)	618.97	26,344.2
	8	0.00965(3.81 × 10^−5^)	15.043(0.49)	614.72	24,470.6
infill percentage	9	0.00770(3.81 × 10^−5^)	16.440(0.24)	490.51	33,516.4
	10	0.010130(4.4 × 10^−5^)	18.010(0.20)	645.51	27,900.8
	11	0.01238(0.00015)	18.645(0.42)	788.84	23,635.9
	12	0.01466(4.40 × 10^−5^)	21.349(0.89)	934.30	22,850.7
	13	0.01635(0.00079)	21.752(0.56)	1041.53	20,885.0
layer height	14	0.00878(2.20 × 10^−5^)	13.223(0.06)	559.52	23,632.9
	15	0.00983(8.80 × 10^−5^)	14.814(0.27)	626.40	23,650.4
	16	0.01065(0.00039)	16.449(0.52)	678.43	24,245.8
orientation	17	0.01073(0.00016)	19.265(0.46)	683.74	28,176.1
	18	0.01127(8.80 × 10^−5^)	14.298(0.66)	717.71	19,922.2
	19	0.00978(9.59 × 10^−5^)	16.482(0.76)	623.22	26,447.1

**Table 3 polymers-14-02446-t003:** Test results for PLA material.

Variable Parameter	Set Number (as in Table 1)	Mean Weight of Set, kg (st. dev.)	Mean UTS, MPa (st. dev.)	Mean Density of Set, kg/m^3^	Mean STS of Set, Nm/kg
temperature	1	0.00923(8.80× 10^−5^)	12.071(0.78)	588.18	20521.9
	2	0.00950(7.62× 10^−5^)	13.342(0.54)	605.17	22047.3
	3	0.00993(4.40× 10^−5^)	15.492(0.22)	632.77	24482.2
	4	0.01015(3.81× 10^−5^)	17.137(0.49)	646.58	26503.7
	5	0.01022(9.59× 10^−5^)	18.429(0.56)	650.82	28316.9
infill type	6	0.00992(2.20× 10^−5^)	15.083(0.49)	631.71	23876.9
	7	0.00997(7.93× 10^−5^)	13.981(0.19)	634.90	22020.4
	8	0.00985(6.60× 10^−5^)	14.256(0.76)	627.47	22720.5
infill percentage	9	0.00753(2.20× 10^−5^)	12.510(0.82)	479.89	26069.2
	10	0.00993(2.20× 10^−5^)	12.978(0.20)	632.77	20509.7
	11	0.01213(5.82× 10^−5^)	14.023(0.08)	772.92	18142.5
	12	0.01482(9.59× 10^−5^)	19.522(0.20)	943.85	20683.3
	13	0.01588(9.59× 10^−5^)	19.949(0.51)	1011.80	19716.3
layer height	14	0.00802(4.40× 10^−5^)	11.470(0.56)	510.68	22460.9
	15	0.00960(3.81× 10^−5^)	12.541(0.02)	611.54	20507.3
	16	0.01075(0.00000)	14.058(0.09)	684.80	20528.7
orientation	17	0.01072(4.40× 10^−5^)	21.699(0.21)	682.67	31785.3
	18	0.01128(2.20× 10^−5^)	9.548(0.15)	718.77	13284.2
	19	0.00992(0.00013)	15.452(0.44)	631.71	24461.0

## Data Availability

The data presented in this study are available upon request from the corresponding author.

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
