# Peer review of "Research on the Influence of Processing Parameters on the Specific Tensile Strength of FDM Additive Manufactured PET-G and PLA Materials"

_polymers, 2022, doi:10.3390/polym14122446_

Round 1

Reviewer 1 Report

The authors present a well set out investigation into the influence of 3D printing parameters on mechanical testing of products.  The manuscript reads well and makes a useful contribution to the literature.  I was initially concerned about how this work is differentiated from previous studies.  A statement in the introduction hints that there has been some similar work on mechanical properties being measured against processing parameters: 

"Previous research on materials used for 3D printing has led to the development of polymers with favorable mechanical properties and biocompatibility, where mechanical properties can be tuned by changing the parameters of the printing process [12]."

However, reference 12 does not go into specifics as this present work does.  There are however relevant studies on some aspects of processing conditions that seem relevant:. A quick search took me to:

1. https://doi.org/10.1080/14658011.2017.1399531,  

2. https://doi.org/10.1080/10426914.2022.2030871

3. Polymers | Free Full-Text | Tensile Mechanical Behaviour of Multi-Polymer Sandwich Structures via Fused Deposition Modelling | HTML (mdpi.com) and I think that there are many more.  As a standalone report, I am confident that this work is sound, but I am concerned about where the originality is; i.e. how is this work sufficiently different from the many other studies of 3D printed materials to merit publication in Polymers?

The experimental description is clear - my only question is whether support structures were used to support the overhang in the XY orientation?

There is some scope for improvement in the discussion of the results, which is very insular.  The authors should remember that most readers with an interest in this area will want to know what it tells them for different materials and a different printer.

For readers of this article, the main interest will be in how the results can be generalised. This research provides a clear technical description of the impact of some procesing parameters with one particular printer and two very specific polymers, but it is not clear if it has value in guiding process parameter for a different PLA, PETG, or even a different material or the same materials printed on a different printer.

This really needs some thought about what is different about the polymer material properties.  E.g. "Variable temperature for PLA sets achieved higher relative increase of both STS and 274 UTS than those made from PET-G" - I want to know why this is in order to know what it might mean for printing in PC or ABS?  Does PLA have a stronger temperature dependence of viscosity or density than PET-G? With some consideration for the underlying reason for the trends/comparison, the article would have much more value.

This is nicely started in teh final line of the discussion: "elongation of PLA in elastic range of deformation for PLA is smaller than for PET-G"  It would be really good to see a side by side comparison of some of the raw elongation data to illustrate this point.  This kind of insight is what would help researchers to know what to look for when considering the polymer science that underpins processing strategy.

Minor points:

[72] ABS is amorphous - it never has crystallinity, so "FDM method doesn't significantly influence the crystallinity degree of ABS," is a bit unnecessary

Figures e.g. fig 6 could be presented a bit better.  The Excel default settings give rather anaemic colours (yellow points dont always print out well), and the equations of best fit would be better expressed in terms of the actual variables (UTS, STS etc , not y and x)

Author Response

Dear Reviewer!

Dear Reviewer! Replies to your comments are included in the file. Thank you!

Authors

Reviewer 2 Report

Effects of printing parameters such as temperature, layer height, infill, etc., on mechanical properties of FDM 3D printed PLA and PET-G were examined in this paper. Please see my comments as below:  

1. The author needs to reorganize the introduction part to make it more concise and straightforward. There is too much irrelevant content in introduction which makes the introduction tedious and distracting. As the topic is effects of processing parameters on mechanical properties of FDM 3D printed PET-G and PLA materials, there is no need to discuss ABS, nylon 6, MJF, DLP, SLA, SLS, etc.

2. SLA in 3D printing refers to stereolithography, instead of “shape memory alloy”

3. The author should explain what are 0- and 45-degree positions when citing the work by Kozior and Kundera.

4. There are incomplete sentences in this manuscript, such as “Three different layer thicknesses (0.15, 0.20, and 0.25 mm), filling degree (30%, 50%, and 70%), and filling structure (rectilinear) [3].” “To improve the quality of prints by this method, the effect of various process parameters such as layer thickness, molding direction, filling rate, filling density [16], printing angles, on mechanical properties including tensile strength and elastic properties of materials used in additive manufacturing methods of FDM [11].”
5. In Figure 3, (b) and (c) are both triangles

6. The author should clarify if the specimen initial cross-section area is the area of the porous dumbbell shape specimen or that of the solid material. Is the area obtained from CAD software calculation or real measurement? The obtained density was acquired by dividing the real mass by theoretical volume, which could bring inaccuracy to density and STS values as the practical amount of material being extruded out from the nozzle can be different from the theoretical value.

7. In Figure 4 and 5, the change trend of temperature, infill percentage and layer height should be marked in the figure.

8. The author needs to have more discussion about why for both materials the optimal value is close to 50 % infill. What are the criteria for optimizing infill? The author states that micro air bubbles might be trapped in PET-G during printing. Did the author run the test a couple more times to avoid bubbles and get the same result? What made the author think that air bubbles were trapped in PET-G whereas PLA did not trap air?

Minor issue:
In the abstract, the “layer height” shows up twice in the sentence “This research presents results of classic tensile strength tests performed on 57 PET-G specimens and 57 PLA specimens manufactured with varying technological parameters such as: printing temperature, print orientation, layer height, infill percentage and layer height.”

Author Response

Dear Reviewer!

Replies to your comments are included in the file. Thank you!

Authors

Round 2

Reviewer 2 Report

This manuscript can be accepted in present form.